

# Strong linkages between surface and deep water dissolved organic matter in the East/Japan Sea

Tae-Hoon Kim[1], Guebuem Kim[2], Yuan Shen[3], Ronald Benner[3,4]

[1] Department of Earth and Marine Sciences, Jeju National University, Jeju, 63243, Republic of Korea
[2] School of Earth and Environmental Sciences/Research Institute of Oceanography, Seoul National University, 1 Gwanak-ro, Gwanak-gu, Seoul 08826, Republic of Korea
[3] Marine Science Program, University of South Carolina, Columbia, South Carolina 29208, USA
[4] Department of Biological Sciences, University of South Carolina, Columbia, South Carolina 29208, USA

*Correspondence to*: Tae-Hoon Kim (thkim@jejunu.ac.kr)

**Abstract.** Vertical and horizontal distributions of total hydrolysable amino acids (THAA), dissolved organic carbon (DOC), and dissolved organic nitrogen (DON) were measured in the East/Japan Sea (EJS). The euphotic zone of this sea is N-limited, and the N:P ratio is ~13 below 200 m depth. Elevated THAA concentrations ($137 \pm 34$ nM) and DOC-normalized yields ($0.8 \pm 0.2\%$ of DOC) were observed in deep waters ($\geq 1000$ m) of the EJS compared with those in the deep North Pacific Ocean.
Significantly high THAA concentrations and yields were observed in a region of deep-water formation, indicating the convection of margin-derived bioavailable dissolved organic matter (DOM) to deep waters. Declining THAA concentrations ($36 \pm 12\%$) and yields ($33 \pm 13\%$) were observed between 1000–3000 m throughout the EJS, indicating the utilization of bioavailable DOM in deep waters. Concentrations of the D-enantiomers of amino acids (Ala, Glx, Asx, and Ser) were relatively high in deep waters of the EJS, indicating substantial bacterial contributions to DOM from surface and upper
mesopelagic waters. These observations suggest that the transport of bioavailable DOM to microbial food webs in deep waters of the EJS is sensitive to changes in deep-water renewal rates.

## 1 Introduction

The East/Japan Sea (EJS) is an enclosed marginal sea in the northwestern Pacific Ocean surrounded by Korea, Japan, and Russia. The EJS consists of three deep basins (>2000 m), including the Ulleung Basin (UB) in the southwest, the Yamato Basin (YB) in the southeast, and the Japan Basin in the northern region. The Tsushima Current (TC) transports warm and saline waters into the EJS through the shallow Korea/Tsushima Strait, which has a sill depth of ~130 m. The TC water
occurs in the upper 150 m of the UB and the YB, and encounters cold waters formed in the northern part of the polar front at about 40°N. The deep water ($\geq$1000 m) in the EJS is formed by deep convection and brine rejection (Martin et al., 1992; Kim et al., 2001; Talley et al., 2003; Postlethwaite et al., 2005; Jenkins, 2008). On the basis of salinity and oxygen-isotope budgets, Postlethwaite et al. (2005) estimated the potential rate of bottom-water formation driven by brine rejection (depth > 2500 m) to be about $4 \times 10^{12}$ m$^3$ yr$^{-1}$, which accounts for 25–35% of the abyssal water formation. Tracer studies have shown
that the turnover time of deep water is on the order of 100 years (Harada and Tsunogai, 1986; Watanabe et al., 1991).





The EJS is a dynamic and productive marginal sea (Jenkins, 2008), with an elevated rate of primary production (~200 g C m$^{-2}$ yr$^{-1}$) based on sediment trap data (Hong, 1998). Surface waters in the EJS are strongly N-limited and dominated by cyanobacteria and pelagophytes (Kim et al., 2010). The concentrations of dissolved organic nitrogen (DON) (4–7 µM) in the

upper 100 m of the EJS are slightly lower than those in major ocean basins due to the apparent uptake of DON by cyanobacteria, as suggested by a strong negative correlation between the concentrations of DON and zeaxanthin, a biomarker of cyanobacteria (Kim and Kim, 2013). Deep waters of the EJS have the highest concentrations (~58 µM) of dissolved organic carbon (DOC) measured in the deep ocean (Kim et al., 2015). Elevated concentrations of DOC in deep waters are indicative of the rapid transport of organic matter from surface waters, but relatively little is known about the

source, reactivity and chemical composition of DOM in the EJS.

Amino acids are major biochemical components of plankton biomass (Wakeham et al., 1997; Kaiser and Benner 2009) and key constituents of marine DOM (Benner, 2002). The abundance and composition of dissolved amino acids can provide insights about ecosystem productivity (Shen et al., 2012; Shen et al., 2016), DOM bioavailability (Davis and Benner 2007),

bacterial contributions to DOM (McCarthy et al. 1998; Kaiser and Benner 2008), and the extent of alteration of organic matter (Cowie and Hedges, 1994; Dauwe and Middelburg, 1998). In the present study, we determined the compositions, including D- and L-enantiomers, and concentrations of total hydrolyzable amino acids (THAA) in the EJS on a basin-wide scale to further evaluate the dynamics of DOC and DON in the EJS.

**2 Materials and Methods**

Seawater samples were collected during two periods: (A) July 9–18, 2009, aboard the R/V M.A. Lavrentyev of the Pacific Oceanological Institute (POI), Russia, and (B) August 8–18, 2009, aboard the R/V Tam-Yang of Pukyung National University (PKNU), Korea (Fig. 1). Seawater samples were filtered onboard through a syringe glass-fiber filter (Whatman,

0.7 µm pore size, 25 mm) for DOC, THAA, TDN (total dissolved nitrogen), and DIN (dissolved inorganic nitrogen; NO$_3^-$ +NO$_2^-$+NH$_4^+$). Water samples for DOC and TDN measurements were acidified with 6 mol L$^{-1}$ HCl to pH ~2 in pre-combusted glass ampoules (550°C for 5 h) and stored at 4°C for preservation until analysis. Samples for DIN and THAA analyses were collected in polyethylene bottles and stored frozen (-20°C) until analysis. Sample handling and preparations were performed in a clean bench (class 100). In the laboratory, the concentrations of DIN were measured using an auto

analyzer (Futura Plus, Alliance Co.) based on the colorimetric analysis. The concentrations of DOC and TDN were measured using a TOC-V$_{CPH}$ analyzer (Shimadzu, Japan) based on high-temperature combustion.

The THAA concentrations were measured using high-performance liquid chromatography (Agilent 1260 with fluorescence detector). The samples were dried with nitrogen gas and hydrolyzed with 6 mol L$^{-1}$ HCl at 150°C for 32.5 min in a CEM



Mars 5000 microwave (Kaiser and Benner 2005). The D- and L-enantiomers of amino acids were derivatized with *o*-phthaldialdehyde and *N*-isobutyryl-L-cysteine (IBLC), and they were separated on a Poroshell 120 EC-C18 column (4.6 × 100 mm, 2.7 μm particles) (Shen et al., in review). Acid-catalyzed racemization of enantiomers during hydrolysis was corrected according to Kaiser and Benner (2005). Eighteen amino acids were included in the analysis: asparagine + aspartic

acid (Asx), glutamine + glutamic acid (Glx), serine (Ser), histidine (His), glycine (Gly), threonine (Thr), β-alanine (β-Ala), arginine (Arg), alanine (Ala), γ-aminobutyric acid (γ-Aba), tyrosine (Tyr), valine (Val), phenylalanine (Phe), isoleucine (Ile), leucine (Leu), and lysine (Lys). The four abundant D-amino acids (D-Asx, D-Glx, D-Ser, and D-Ala) are reported in this study.

The concentrations of DON were determined indirectly by subtracting the concentrations of DIN (nitrate + nitrite +
ammonium) from concentrations of TDN. This calculation can produce large uncertainties when the total concentrations of dissolved nutrients are similar to those of dissolved inorganic nutrients. However, the average concentrations of TDN in surface and deep waters (15.6 ± 5.8 and 26.6 ± 2.4 μM) were much higher than those of DIN (10.8 ± 6.3 and 23.7 ± 2.6 μM), thus allowing a reasonable comparison of the overall depth trends of DON.

**3 Results and Discussion**

**3.1 Concentrations and distributions of DOM and THAA in the EJS**

The concentrations of DOC and DON ranged from 57 to 83 μM and 2.0 to 8.4 μM, respectively, in the EJS (Table 1; Fig. 2).
The average DOC and DON concentrations were 72 ± 6 μM and 4.8 ± 0.8 μM in surface waters (0–200 m) and 59 ± 1 μM and 2.6 ± 0.3 μM in deep waters (≥ 1000 m), respectively. The concentrations of DOC and DON in deep waters were approximately 18% and 54% lower than those in surface waters, indicating the preferential remineralization of DON in deep waters. Deep-water concentrations of DOC in the EJS were higher than those in the major oceans (34–48 μM) (Bauer et al., 1992; Sharp et al., 1995; Thomas et al., 1995; Hansell and Carlson, 1998). Kim et al. (2015) suggested the high
concentrations of DOC in deep waters of the EJS are due to low rates of degradation (0.04 μmol C kg$^{-1}$ yr$^{-1}$) associated with low water temperatures (<1°C).

The concentrations of DON in surface waters of the EJS were similar to those in the global surface ocean (4.4 ± 0.5 μM) (Letscher et al., 2013). Unlike DOC, the concentrations of DON in deep waters of the EJS were similar to values in the
30 Eastern Pacific (2.5 ± 0.4 μM) and slightly lower than those in the Southern Ocean (3.5 ± 0.6 μM), the North Atlantic (3.2 ± 0.3 μM), and the Mediterranean Sea (3.5 ± 0.4 μM) (Loh and Bauer, 2000; Hansell and Carlson, 2001; Pujo-Pay et al., 2011). The DOC:DON ratios in deep waters of the EJS (23 ± 3) were higher than those in the North Pacific, North Atlantic, and Mediterranean Sea (Loh and Bauer, 2000; Hansell and Carlson, 2001; Pujo-Pay et al., 2011) (Fig. 2).





Kim and Kim (2013) found the DOC:DON ratios in the surface EJS were similar to those in the North Pacific Ocean, which is N-limited (Zehr et al., 2001). In the EJS cyanobacteria dominate (20–65%) the phytoplankton community in the surface mixed layer, where N:P ratios were <5 (Kim et al., 2010). Significant negative correlations between the concentrations of zeaxanthin, a marker pigment of cyanobacteria, and DON concentrations have been observed in the surface mixed layer of

the EJS, indicating that C:N stoichiometry of DOM can be strongly influenced by phytoplankton (Kim and Kim, 2013).

The concentrations of total hydrolysable amino acids (THAA) ranged from 87 to 290 nM in the EJS (Table 1; Fig. 2). The average THAA concentrations decreased by ~37% from 248 ± 27 nM in surface waters to 137 ± 34 nM in deep waters (Fig. 3). The concentrations of THAA in the EJS were in general higher than those at the Bermuda Atlantic Time-series Study

(BATS) site (150–198 µM in surface waters; 87–103 µM in deep waters) and the Hawaii Ocean Time-series (HOT) site (190–285 µM in surface waters; 45–70 µM in deep waters) (Fig. 4; Kaiser and Benner, 2008; Kaiser and Benner, 2009).

Abyssal circulation in the EJS has relatively strong cyclonic flows along the basin periphery and sluggish flows in the interior region (Fig. 1; Senjyu et al., 2005). In this context, the concentrations of DOC in the deep layer of the EJS decreased

from the north (59 ± 3 µM) to south (55 ± 2 µM) along the abyssal circulation, with a similar trend of AOU (apparent oxygen utilization) (Kim et al., 2015). The lowest concentration of THAA (87 nM) was found at 2000 m depth in the southern frontal zone (station 7). This station is located in the central Ulleung Basin, which may have the oldest water among the sampling sites. The deep water in the Ulleung Basin is known to originate from the northern Japan Basin (Kim et al., 1991; Senjyu et al., 2005).

### 3.2 Bioavailable DOM in the EJS

The amino acid components of DOM comprise an important fraction of the bioavailable DOM in seawater, and the DOC-normalized yield of THAA is a useful proxy of bioavailable DOM (Davis and Benner, 2007). THAA accounted for 0.5 to 1.5%

of the DOC and 4.1 to 9.1% of the DON in the EJS (Fig. 3). THAA yields were 1.2 ± 0.1% of the DOC and 6.7 ± 1.2% of the DON in surface waters, and 0.8 ± 0.2% of the DOC and 6.4 ± 1.1% of the DON in deep waters (Table 1). The yields of amino acids (%DOC) decreased from surface to deep waters (Fig. 3), indicating the components of DOM containing amino acids were more bioavailable than bulk DOC. The average yields of THAA in deep waters of the EJS were similar to those (0.7 ± 0.1%DOC) at the BATS site and were greater than those (0.5 ± 0.1%DOC) at the HOT site (Kaiser and Benner, 2009),

indicating the occurrence of some bioavailable DOM in EJS and BATS deep waters (Fig. 4).

Yields (%DOC) of THAA decreased by 33 ± 13% in deep waters (1000–3000 m). The declines in THAA yields (%DOC) in the EJS were greater than those at BATS (25%) and HOT (17%) (Kaiser and Benner, 2009), indicating a greater reactivity of



THAA in deep waters of the EJS. The yields of THAA in bottom waters of the EJS were below 0.7 %DOC, indicating the refractory nature of DOM in bottom waters with ventilation ages of ~100 years (Harada et al., 1986; Watanabe et al., 1991).

There was considerable spatial variability in THAA yields (%DOC) and bioavailable DOM among stations in the EJS (Fig.
3). This variability was particularly evident in deep waters at station 2, which had very high THAA yields (1.3 and 1.2 %DOC) at 1000 and 1500 m, indicating the presence of bioavailable DOM with a semi-labile nature (Davis and Benner, 2007). These high yields at depth are consistent with deep-water convection in this region of the EJS and the potential input of DOM from nearby margin surface waters (Kim et al., 2002; Talley et al., 2003). In contrast, the low THAA yield (0.6 %DOC) at 3000 m was indicative of DOM with a refractory nature (Davis and Benner, 2007). Deep convection in the
EJS appears to have weakened in recent decades in response to climate change (Gamo, 2011), and the low THAA yields and refractory nature of DOM in abyssal waters (≥2500 m) are consistent with longer residence times and a greater extent of microbial decomposition.

### 3.3 Amino acid degradation index (DI) and non-protein amino acids in the EJS

The amino acid degradation index (DI), which is based on the composition of protein amino acids, and the mol % of the non-protein amino acids (β-Ala + γ-Aba) have been used as indicators of biogeochemical alterations of DOM (Davis et al., 2009). The DI values ranged from -2.33 to 0.70 in the EJS, with more negative values indicating a greater extent of alteration (Table 1). The average DI values in the EJS were -0.93 ± 0.74 in surface water and -0.72 ± 0.62 in deep water. The DI values
decreased with depth at the BATS and HOT sites (Kaiser and Benner, 2009), whereas those in the EJS were more variable. The average DI values in the EJS deep waters were intermediate between those (-1.40 ± 0.41) at HOT and those (0.74 ± 0.69) at BATS (Kaiser and Benner, 2009) (Fig. 4).

The mole percentages (mol %) of β-Ala and γ-Aba ranged from 1 to 9 mol % and 0 to 3 mol %, respectively, in the EJS
(Table 1). The average β-Ala and γ-Aba mole percentages were 5 ± 1 mol % and 1 ± 1 mol % in surface waters and 5 ± 1 mol % and 2 ± 1 mol % in deep waters, respectively. The average β-Ala + γ-Aba mole percentages in surface waters were similar (±10%) to those in the deep water (Fig. 4). The mole percentages of β-Ala + γ-Aba in surface and deep waters of the EJS were lower than those at BATS and HOT (Kaiser and Benner, 2009). The mole percentages of β-Ala in the EJS were similar to those at BATS and HOT, while the mole percentages of γ-Aba was lower than those at BATS and HOT (Kaiser
and Benner, 2009) (Fig. 4).

The mol % of β-Ala + γ-Aba increased with depth at the HOT site (Kaiser and Benner, 2009), while those in the EJS were variable with depth as was observed with the DI. In this study, there is a significant, but weak, correlation between the amino





acid yield (%DOC) and the DI. Similar observations were made in the Western Arctic Ocean (Shen et al., 2012). The DI was negatively correlated ($r^2$=0.395) with the mol % of β-Ala + γ-Aba in the EJS. Variable DI values in the Chukchi and Beaufort Seas appeared to be associated with riverine sources (Shen et al., 2012). Davis et al. (2009) found that the DI and the mol % of β-Ala + γ-Aba were only effective indicators of DOM alteration during the later stages of decomposition. It

appears the high variability in the DI and mol % of β-Ala + γ-Aba is consistent with the dynamic nature of this marginal sea.

**3.4 Bacterial contributions to DOM in the EJS**

The D-enantiomers of four amino acids (Ala, Glx, Asx, and Ser) are biomarkers of the bacterial origin of amino acids and

DOM (McCarthy et al., 1998; Kaiser and Benner, 2008). Of these amino acids, D-Ala and D-Asx concentrations were higher than those of D-Glx and D-Ser, and the concentrations of the four L-enantiomers were higher than those of the respective D-enantiomers in the EJS (Fig. 5 and Table 2). Similar patterns were observed at the BATS and HOT sites (Kaiser and Benner, 2008; Kaiser and Benner, 2009), but, the concentrations of D-amino acids in deep waters of the EJS were higher than those at the BATS and HOT sites. The elevated concentrations of D-amino acids in deep waters are consistent with previous

observations of the accumulation of marine humic-like fluorescent DOM in deep waters of the EJS (Kim and Kim, 2015; Kim and Kim, 2016).

Proteins are comprised of L-amino acids, whereas most D-amino acids reside in a variety of non-protein biomolecules associated with the cell wall-membrane complex of bacteria (Schleifer and Kandler, 1972; Kaiser and Benner, 2008).

Bioassay studies indicate D-amino acids are generally more resistant to decomposition than L-amino acids, so the D:L ratios of Ala, Glx, Asx, and Ser increase rapidly during bacterial growth and utilization of organic matter (Jørgensen et al., 1999; Amon et al., 2001; Kawasaki and Benner, 2006). The average D:L ratios in surface waters (0.73 ± 0.22 Asx, 0.26 ± 0.10 Glx, 0.38 ± 0.19 Ser, 0.65 ± 0.18 Ala) and deep waters (0.61 ± 0.21 Asx, 0.23 ± 0.10 Glx, 0.37 ± 0.26 Ser, 0.63 ± 0.23 Ala) of the EJS were similar, with a slight decrease observed at depth (Fig. 5 and Table 2). The elevated D:L ratios in the EJS are

indicative of the rapid utilization of proteinaceous material and the accumulation of D-amino acids derived from bacteria.

Surface waters of the EJS have low N:P ratios (<5) and low DIN concentrations (<1 μM) in the mixed layer, and they are often dominated by cyanobacteria (Kim et al., 2010; Kim and Kim, 2013). It appears that cyanobacteria (*Trichodesmium* sp. and *Synechococcus* sp.) produce D-Ala and D-Glx, but not D-Asx or D-Ser (Kaiser and Benner, 2008). In addition to D-Ala

and D-Glx, many heterotrophic marine bacteria also produced D-Asx and D-Ser (Kaiser and Benner, 2008). The elevated D:L ratios of these four biomarker amino acids in the EJS are consistent with major contributions from heterotrophic bacteria, but they do not preclude contributions from $N_2$-fixing cyanobacteria, such as *Trichodesmium* spp., in surface waters.



The D:L ratios of Asx and Ala, were somewhat elevated in surface waters of the EJS compared with those at the BATS and HOT sites, whereas the D:L ratios of Glx and Ser were elevated in deep waters at the HOT site (Fig. 6; Kaiser and Benner, 2008). The high D:L ratios of these biomarker amino acids throughout the water column of the EJS and the HOT and BATS sites indicate bacteria are an important source of amino acids, DOC and DON in the global ocean (Kaiser and Benner, 2008; Benner and Herndl, 2011).

### 3.5 Implications

The recent slowdown of the deep-water formation in the EJS has been well documented using the change in dissolved oxygen concentrations in deep water masses (Kim et al., 2001). As such, the rate of anthropogenic $CO_2$ accumulation in the deep layer of the EJS has been decreasing considerably for the last few decades (1992~2007) owing to the shallowing of deep-water formation (Park et al., 2008). It is unknown whether this process also impacts bioavailable DOM accumulation in the deep EJS. However, our results suggest that the shallowing or slowdown of deep-water formation in response to atmospheric warming would bring about a considerable decrease in the oceanic storage of bioavailable DOM and a consequent positive feedback in the climate system.

### Acknowledgements

We would like to thank the captain, crew and EAST-I participants for assistance with the field sampling. This research was funded by the Ministry of Oceans and Fisheries, Korea, through the project titled "East Asian Seas Time series-I (EAST-I) (20110093)" and by the National Research Foundation (NRF) of Korea (NRF-2016R1C1B2006774). RB and YA acknowledge support from National Science Foundation (USA) grant 1233373.

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





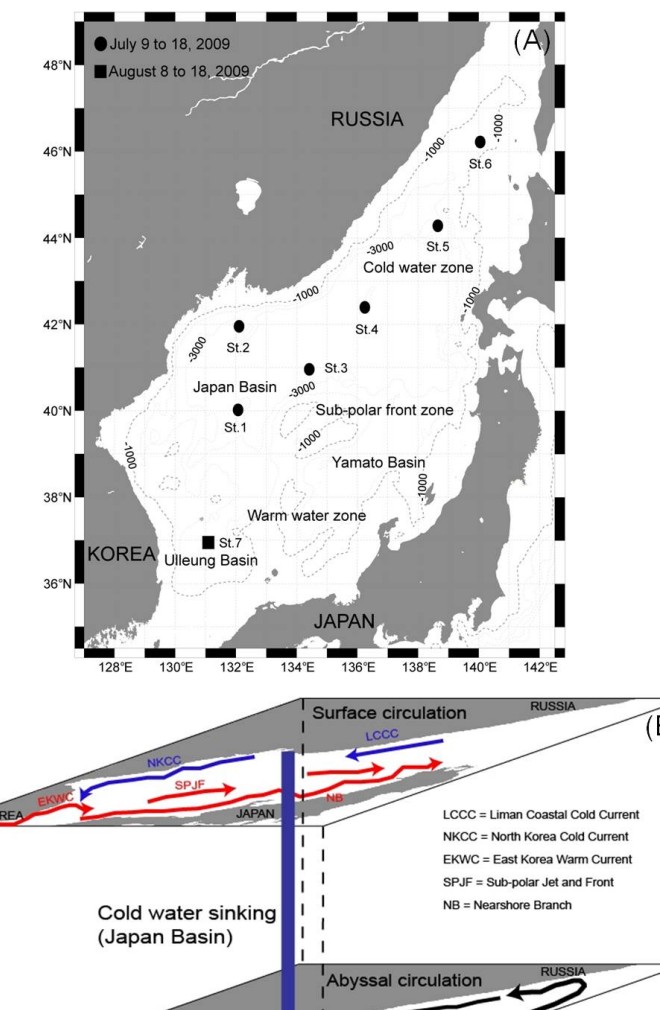

**Figure 1: (A) A map of the region showing stations (St.) in the East/Japan Sea. (B) A schematic of the water circulations of the surface and bottom waters in the East/Japan Sea (Senjyu et al., 2005).**





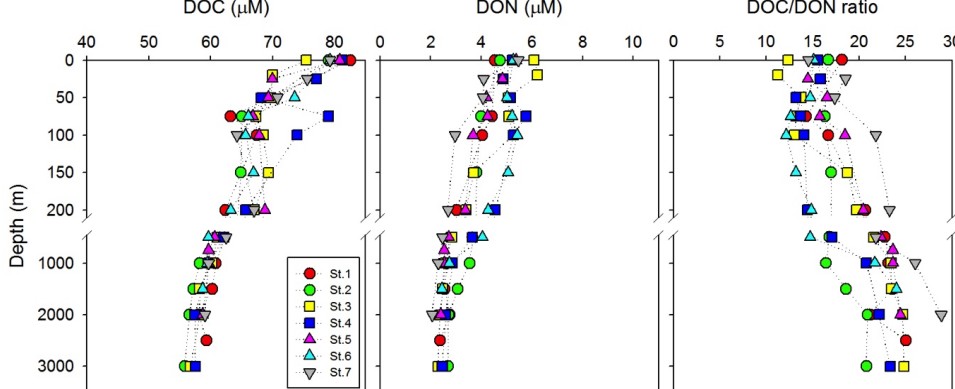

**Figure 2: Vertical profiles of dissolved organic carbon (DOC), dissolved organic nitrogen (DON), and the DOC:DON ratio in the East/Japan Sea.**





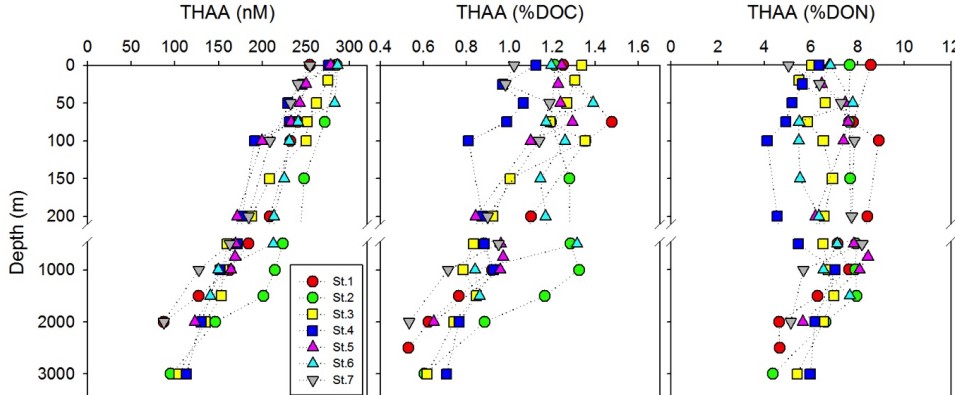

**Figure 3: Vertical profiles of total hydrolysable amino acids (THAA), DOC-normalized yields of THAA (%DOC), DON-normalized yields of THAA (%DON), in the East/Japan Sea.**





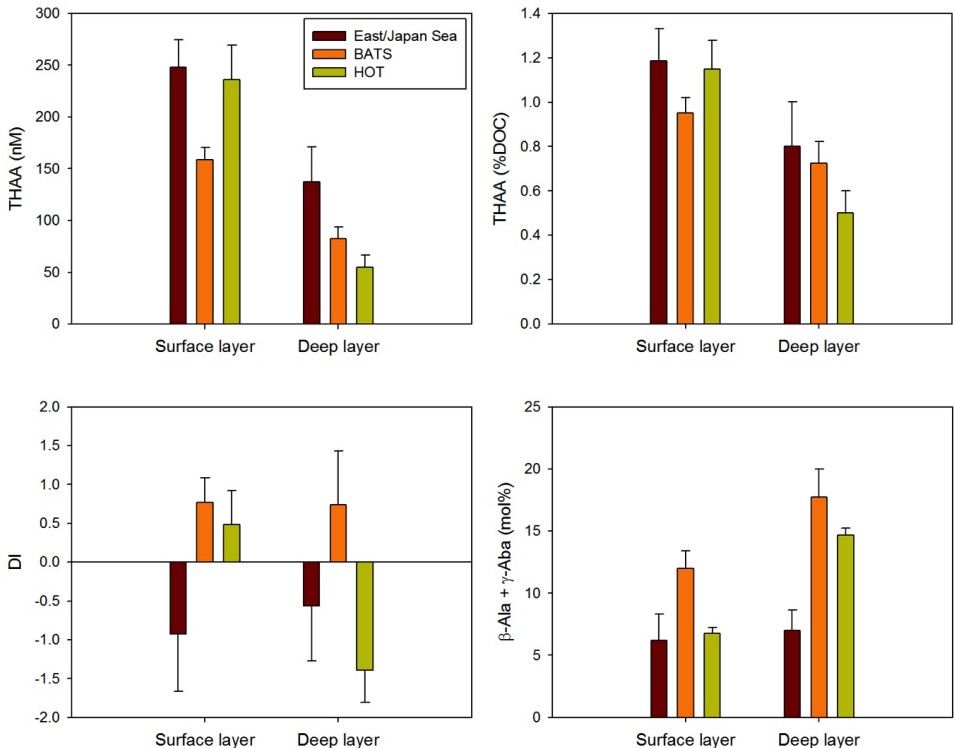

**Figure 4: Average total hydrolysable amino acids (THAA) concentrations, DOC-normalized yields of THAA (%DOC), degradation index, and mole percentage (mol%) of β-Ala + γ-Aba from surface (0–200 m) and deep (≥ 1000 m) layers in the East/Japan Sea, BATS, and HOT. Error bars represent standard deviations.**





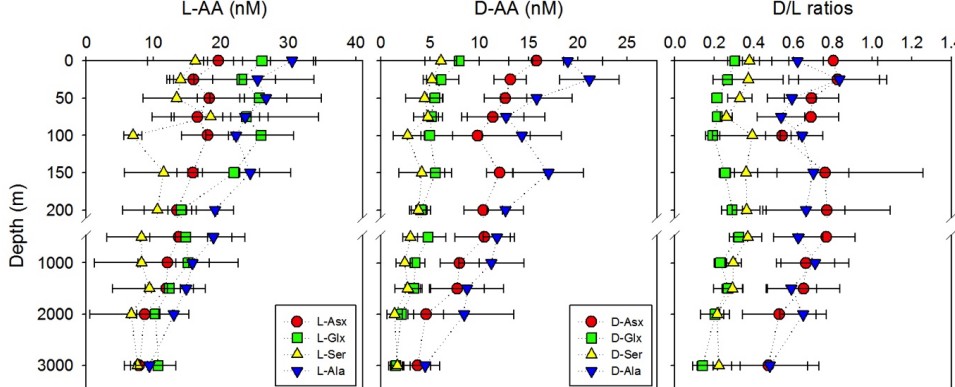

**Figure 5: Average concentrations of total hydrolysable L- and D- enantiomers of asparagine + aspartic acid (Asx), glutamine + glutamic acid (Glx), serine (Ser), and alanine (Ala) concentrations in the EJS, and the average D:L ratios for these amino acids.**



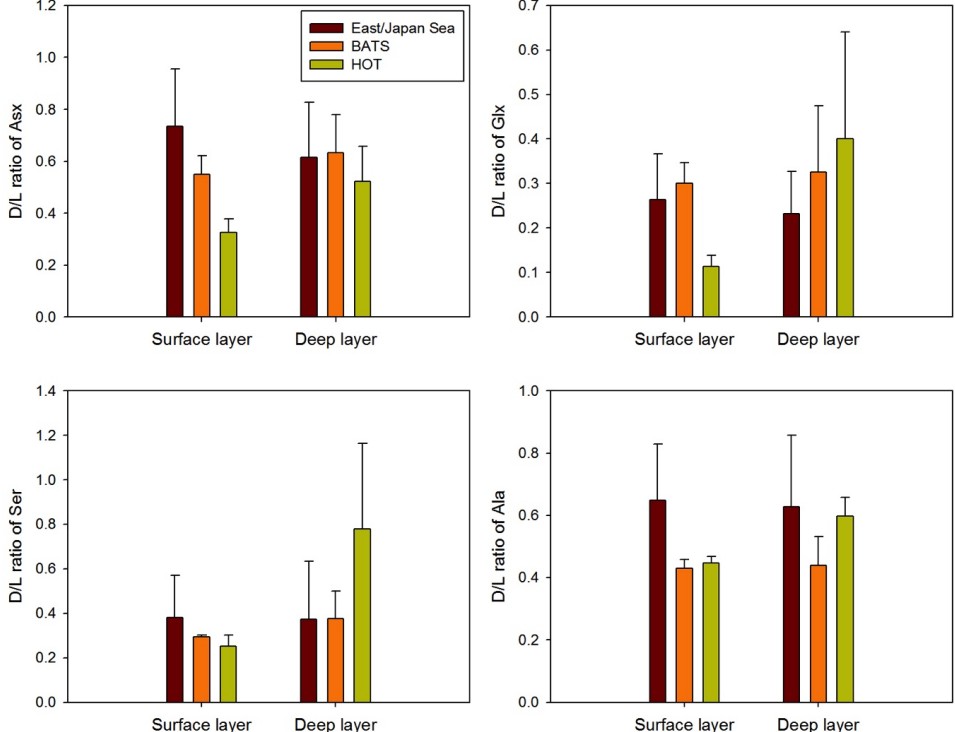

**Figure 6: Average D:L ratios of asparagine + aspartic acid (Asx), glutamine + glutamic acid (Glx), serine (Ser), and alanine (Ala) from surface (0–200 m) and deep (≥ 1000 m) layers in the East/Japan Sea, BATS, and HOT. Error bars represent standard deviations.**



Table 1. Physicochemical characteristics in the East/Japan Sea.

| Station | Depth (m) | Temp. (°C) | Sal. (psu) | DOC (μM) | DON (μM) | THAA (nM) | THAA (%DOC) | DI | β-Ala (mol%) | γ-Aba (mol%) |
|---|---|---|---|---|---|---|---|---|---|---|
| 1 | 0 | 21.079 | 33.734 | 83 | 4.5 | 290 | 1.2 | -0.87 | 3 | 1 |
| | 75 | 10.481 | 34.164 | 63 | 4.4 | 263 | 1.5 | -0.24 | 3 | 1 |
| | 100 | 10.323 | 34.177 | 67 | 4.0 | 234 | 1.4 | 0.70 | 1 | 0 |
| | 200 | 8.688 | 34.188 | 62 | 3.0 | 203 | 1.1 | -0.78 | 5 | 1 |
| | 500 | 0.764 | 34.060 | 62 | 2.7 | 181 | 0.9 | -1.44 | 9 | 2 |
| | 1000 | 0.264 | 34.064 | 61 | 2.6 | 170 | 0.9 | -1.06 | 5 | 2 |
| | 1500 | 0.154 | 34.064 | 60 | 2.6 | 132 | 0.8 | -0.07 | 5 | 1 |
| | 2000 | 0.102 | 34.064 | 58 | 2.7 | 96 | 0.6 | 0.70 | 4 | 1 |
| | 2500 | 0.086 | 34.065 | 59 | 2.4 | 90 | 0.5 | -0.38 | 4 | 2 |
| 2 | 0 | 21.079 | 33.734 | 79 | 4.7 | 285 | 1.2 | -1.19 | 6 | 1 |
| | 75 | 10.481 | 34.164 | 65 | 4.0 | 272 | 1.2 | -2.33 | 8 | 1 |
| | 150 | 10.323 | 34.177 | 65 | 3.8 | 248 | 1.3 | -1.16 | 5 | 1 |
| | 500 | 8.688 | 34.188 | 61 | 3.7 | 224 | 1.3 | -0.52 | 5 | 2 |
| | 1000 | 0.764 | 34.060 | 58 | 3.5 | 214 | 1.3 | -0.28 | 3 | 1 |
| | 1500 | 0.264 | 34.064 | 57 | 3.1 | 202 | 1.2 | -1.00 | 6 | 2 |
| | 2000 | 0.154 | 34.064 | 57 | 2.7 | 146 | 0.9 | -0.82 | 4 | 1 |
| | 3000 | 0.102 | 34.064 | 56 | 2.7 | 95 | 0.6 | 0.05 | 5 | 1 |
| 3 | 0 | 18.044 | 33.814 | 75 | 6.1 | 277 | 1.3 | -0.30 | 3 | 1 |
| | 20 | 6.882 | 34.061 | 70 | 6.2 | 275 | 1.3 | -1.24 | 6 | 2 |
| | 50 | 4.199 | 34.072 | 70 | 5.1 | 262 | 1.3 | -1.10 | 5 | 1 |
| | 75 | 3.430 | 34.075 | 67 | 5.1 | 252 | 1.2 | -1.49 | 5 | 3 |
| | 100 | 3.187 | 34.072 | 69 | 5.3 | 250 | 1.4 | -0.21 | 4 | 1 |
| | 150 | 2.666 | 34.063 | 69 | 3.7 | 208 | 1.0 | -1.17 | 6 | 2 |
| | 200 | 2.071 | 34.059 | 67 | 3.4 | 188 | 0.9 | -1.38 | 5 | 1 |
| | 500 | 0.949 | 34.070 | 61 | 2.8 | 159 | 0.8 | -1.45 | 6 | 2 |
| | 1000 | 0.390 | 34.068 | 60 | 2.6 | 155 | 0.8 | -2.02 | 8 | 3 |
| | 1500 | 0.186 | 34.065 | 58 | 2.5 | 153 | 0.8 | -1.03 | 7 | 2 |
| | 2000 | 0.115 | 34.065 | 57 | 2.4 | 135 | 0.7 | -1.62 | 7 | 2 |
| | 3000 | 0.082 | 34.066 | 57 | 2.3 | 105 | 0.6 | -0.73 | 6 | 1 |
| 4 | 0 | 15.841 | 33.599 | 81 | 5.2 | 276 | 1.1 | -1.36 | 5 | 1 |
| | 25 | 5.904 | 34.013 | 77 | 4.9 | 246 | 1.0 | -2.03 | 8 | 2 |
| | 50 | 3.533 | 34.087 | 68 | 5.2 | 229 | 1.1 | -1.62 | 5 | 2 |
| | 75 | 2.458 | 34.087 | 79 | 5.8 | 231 | 1.0 | -0.65 | 6 | 1 |
| | 100 | 2.089 | 34.086 | 74 | 5.3 | 191 | 0.8 | -2.00 | 7 | 2 |
| | 200 | 1.252 | 34.068 | 66 | 4.6 | 177 | 0.9 | -1.35 | 6 | 1 |
| | 500 | 0.617 | 34.068 | 62 | 3.6 | 171 | 0.9 | -1.18 | 7 | 2 |
| | 1000 | 0.258 | 34.066 | 60 | 2.9 | 152 | 0.9 | 0.02 | 4 | 1 |
| | 2000 | 0.100 | 34.065 | 57 | 2.6 | 130 | 0.8 | -0.30 | 6 | 1 |
| | 3000 | 0.081 | 34.066 | 57 | 2.5 | 113 | 0.7 | 0.12 | 4 | 1 |
| 5 | 0 | 14.574 | 33.574 | 81 | 5.4 | 278 | 1.2 | -0.46 | 3 | 1 |
| | 25 | 3.625 | 34.069 | 70 | 4.8 | 250 | 1.2 | -1.26 | 4 | 1 |
| | 50 | 2.238 | 34.080 | 69 | 4.2 | 243 | 1.2 | -0.72 | 4 | 1 |
| | 75 | 1.705 | 34.067 | 67 | 4.2 | 233 | 1.3 | 0.03 | 2 | 0 |
| | 100 | 1.320 | 34.066 | 68 | 3.7 | 200 | 1.1 | 0.23 | 5 | 1 |
| | 200 | 0.881 | 34.065 | 69 | 3.4 | 171 | 0.8 | -0.73 | 6 | 1 |
| | 500 | 0.507 | 34.066 | 61 | 2.7 | 169 | 1.0 | -0.81 | 5 | 1 |
| | 750 | 0.320 | 34.066 | 60 | 2.5 | 169 | 1.0 | -0.77 | 5 | 1 |
| | 1000 | 0.219 | 34.065 | 60 | 2.5 | 164 | 1.0 | -0.56 | 5 | 1 |
| | 2000 | 0.106 | 34.065 | 59 | 2.4 | 123 | 0.6 | -1.85 | 8 | 3 |



| Station | Depth (m) | Temp. (°C) | Sal. (psu) | DOC (μM) | DON (μM) | THAA (nM) | THAA (%DOC) | DI | β-Ala (mol%) | γ-Aba (mol%) |
|---|---|---|---|---|---|---|---|---|---|---|
| 6 | 0 | 12.608 | 33.498 | 79 | 5.3 | 286 | 1.2 | -0.73 | 6 | 1 |
| | 50 | 3.218 | 34.085 | 74 | 5.0 | 283 | 1.4 | -1.34 | 3 | 1 |
| | 75 | 2.227 | 34.076 | 66 | 5.2 | 241 | 1.2 | -0.36 | 7 | 1 |
| | 100 | 1.887 | 34.070 | 66 | 5.4 | 231 | 1.3 | -1.35 | 3 | 1 |
| | 150 | 1.505 | 34.075 | 67 | 5.1 | 225 | 1.1 | 0.10 | 5 | 3 |
| | 200 | 1.305 | 34.080 | 63 | 4.3 | 213 | 1.2 | -0.76 | 4 | 1 |
| | 500 | 0.568 | 34.067 | 60 | 4.0 | 212 | 1.3 | -0.71 | 4 | 1 |
| | 1000 | 0.233 | 34.065 | 59 | 2.7 | 149 | 0.8 | -0.07 | 6 | 1 |
| | 1500 | 0.137 | 34.065 | 59 | 2.4 | 141 | 0.9 | -0.01 | 5 | 1 |
| 7 | 0 | 22.545 | 33.705 | 79 | 5.4 | 254 | 1.0 | -1.58 | 6 | 2 |
| | 25 | 20.677 | 33.810 | 76 | 4.1 | 241 | 1.0 | -1.92 | 6 | 2 |
| | 50 | 17.306 | 34.241 | 71 | 4.1 | 232 | 1.2 | -0.02 | 4 | 1 |
| | 100 | 14.636 | 34.151 | 64 | 2.9 | 209 | 1.1 | -1.07 | 5 | 1 |
| | 200 | 1.989 | 34.050 | 67 | 2.9 | 185 | 0.9 | -1.23 | 7 | 1 |
| | 500 | 0.532 | 34.065 | 62 | 2.9 | 163 | 0.9 | -0.35 | 5 | 1 |
| | 1000 | 0.242 | 34.064 | 60 | 2.3 | 127 | 0.7 | -1.07 | 6 | 1 |
| | 2000 | 0.097 | 34.065 | 59 | 2.0 | 87 | 0.5 | 0.10 | 5 | 2 |



Table 2. Average concentrations of D- and L-enantiomers of aspartic acid and asparagines (D- and L-Asx), glutamic acid and glutamine (D- and L-Glx), serine (D- and L-Ser), and alanine (D- and L-Ala) and the enantiomeric D:L ratios.

| Depth (m) | D-Asx | L-Asx | D-Glx | L-Glx (nM) | D-Ser | L-Ser | D-Ala | L-Ala | D:L-Asx | D:L-Glx | D:L-Ser | D:L-Ala |
|---|---|---|---|---|---|---|---|---|---|---|---|---|
| 0 | 16 | 20 | 8 | 26 | 6 | 16 | 19 | 31 | 0.80 | 0.31 | 0.38 | 0.62 |
| 25 | 13 | 16 | 6 | 23 | 5 | 14 | 21 | 25 | 0.83 | 0.27 | 0.37 | 0.83 |
| 50 | 13 | 18 | 5 | 26 | 4 | 13 | 16 | 27 | 0.69 | 0.21 | 0.33 | 0.59 |
| 75 | 11 | 16 | 5 | 24 | 5 | 18 | 13 | 24 | 0.69 | 0.22 | 0.26 | 0.54 |
| 100 | 10 | 18 | 5 | 26 | 3 | 7 | 14 | 22 | 0.54 | 0.19 | 0.39 | 0.64 |
| 150 | 12 | 16 | 6 | 22 | 4 | 11 | 17 | 24 | 0.76 | 0.25 | 0.36 | 0.70 |
| 200 | 10 | 14 | 4 | 14 | 4 | 11 | 13 | 19 | 0.77 | 0.29 | 0.36 | 0.66 |
| 500 | 11 | 14 | 5 | 15 | 3 | 8 | 12 | 19 | 0.77 | 0.32 | 0.37 | 0.63 |
| 1000 | 8 | 12 | 4 | 15 | 2 | 8 | 11 | 16 | 0.66 | 0.23 | 0.30 | 0.71 |
| 1500 | 8 | 12 | 3 | 12 | 3 | 9 | 9 | 15 | 0.65 | 0.27 | 0.29 | 0.59 |
| 2000 | 5 | 9 | 2 | 10 | 1 | 7 | 8 | 13 | 0.53 | 0.20 | 0.21 | 0.65 |
| 3000 | 4 | 8 | 2 | 11 | 2 | 8 | 5 | 9 | 0.47 | 0.14 | 0.22 | 0.48 |