# Peer review of "Strong linkages between surface and deep water dissolved organic matter in the East/Japan Sea"

_Biogeosciences, 2017_

## Referee Comment (RC1) · J. Hawkes (Referee) · 24 Feb 2017

This article provides new bulk concentration data on amino acids, DOC and DON in the East/Japan Sea. The results are compared with the BATS and HOT oceanographic stations, which provides a useful reference for the relative levels of the various analytes measured. The data support recent findings that suggest that DOM is transported to the deep EJS in higher concentrations than is oceanographically typical, which is an interesting result and makes this a fascinating study region. The data are of a good quality and the discussion is appropriate. Some aspects need to be clarified before publication in Biogeosciences.

Major comments: -Page 2 line 30: There is no mention of the use of reference materials (e.g. the seawater reference material from Hansell's lab). If no references were used,

this should be mentioned. Maybe the results can be compared with another study from the same waters.

- Page 4 Line 15 and many other places: When values are stated as  $\pm$  something, are they referring to one standard deviation? This should be stated (i.e. (mean  $\pm$  SD)). If this is the case, I don't think the values are significantly different on page 4 line 15

-Page 4 Line 31 onwards: They talk about the reactivity of THAAs in the EJS compared with BATS and HOT, but there is no indication of how long a time passes between the sampling points in each case. A comparison of decrease in concentration alone cannot be used to talk about reactivity, without time. This needs an extra paragraph of discussion.

-Page 5 Line 31 onwards: They mention a 'significant but weak' correlation between two variables without providing any statistical data or graphs, only a table. I think this sentence should include at least an r2, p value and n.

-Page 6 lines 9-25: They state that D-amino acids are more resistant to decomposition, yet the ratio of D:L is lower in the deep water than surface water. These two things don't seem to connect, please clarify.

-Section 3.5: I don't understand how they make the step from the reported data to the statement: "our results suggest that the shallowing or slowdown of deep-water formation in response to atmospheric warming would bring about a considerable decrease in the oceanic storage of bioavailable DOM and a consequent positive feedback in the climate system". This needs to be discussed more thoroughly

General comment: There is a lot of discussion of the trends in shallow vs. deep waters, but very little discussion of the lateral data between stations (i.e. figures 2,3 and 5). Some of the stations seem to be significantly different from each other and I would like to see a little discussion on whether these changes fit with the other trends, for example degradation of L-amino acids in comparison to D- or loss of DON preferentially
over DOC. What is the circulation time difference between the different stations in deep water?

Minor comments:

- Page 1 Line 26: Should this read 'surrounded by the Korean Peninsula...?' - The Authors presumably know better than I.

- Page 1 Line 29: The Korea/Tsushima Strait is not indicated on the map, which would be useful

- Page 2 Line 30: 'Based on colorimetric analysis' (no 'the'). Is it possible to reference the method used here?

-Page 3 Line 22-23: This implicitly assumes that all organic matter in these deep waters comes from surface waters. Can this be substantiated?

-Page 4 Lines 10-11: Amino acids are stated in  $\mu$ M, I think this should be nM.

-Page 4 Line 24: I don't think the 'normalized yield of THAA' is a commonly understood concept, so it should be briefly explained here.

BGD

---

## Referee Comment (RC2) · Anonymous Referee #2 · 8 Mar 2017

The manuscript by Kim et al. describes concentrations of dissolved organic nitrogen, dissolved organic carbon and amino acids in the East Japan Sea. A reasonable amount of new data is presented and the presentation is generally aceptable.

The manuscript is well-written, and the organization is easy and clear to follow. It is a short paper that fits within the scope of the journal. The topic is relevant given the possible effects of changes in oceanic carbon chemistry on bacterioplankton communities and subsequent biogeochemical nutrient cycles. The article suggest an interesting link between microbial metabolism and oceanographic processes, it contributes to improve our understanding on the carbon cycle of one of the most understudied variables, so I consider that this manuscript would be suitable for publication after minor revisions.

Major comments:

There are no definition of THAA, DOC or DON in a explicit way. DOC and DON are maybe more obvious, but a short sentence with the main meaning of THAA would be appreciated.

Material and Methods should include info about error of the measurements. Same for the Tables 1 and 2. Certified Reference Materials are widely used nowadays for measurement validation, if they were not used should be said.

Along the manuscript, decreasing concentrations are always related with utilization. What about the lateral advection? Don't should be discussed?

Specific comments:

- Fig.1 - why is not Tsushima Current (TC) in the figure? Additionally, some color differentiation would be appreciated for the map A).

- Page 5 - Line 32 " a significant, but weak, correlation" more info is needed

- Page 6 Line 24 - Is the slight decrease significant?

---

## Referee Comment (RC3) · Anonymous Referee #3 · 9 Mar 2017

Major Comments: This study documents dissolved organic carbon (DOC) and total hydrolysable amino acid (THAA) concentrations in the East Japan Sea, and compares these with values from the North Atlantic and North Pacific. The East Japan Sea is a region of deep water formation, and the study describes elevated THAA concentrations in deep waters that they interpret result from export from surface waters. They also assume that the export of THAAs from surface to depth as a source of bioavailable DOM to deep waters. The study also examines ratios of amino acids to calculate a degradation index proxy for degradation of dissolved organic matter. They find little to no increase in the degradation index with depth, and the absolute degradation index is intermediate between values observed at the HOT and BATS locations.

The manuscript is clear and straightforward. The data are largely consistent with prior work on marine DOM, and the conclusions largely follow from the data. My most

significant concern is that the concluding sentence, regarding the potential decrease in DOM sequestration in the deep ocean with decreasing rates of deep-water formation, is not supported by the observations; this study did not document rates of deep-water formation nor how DOC concentrations have changed through time. The degradation index or %THAA are not precise enough metrics to be able to draw this conclusion.

Section 3.2, lines 23-30: It's not clear that THAA's were preferentially consumed with depth, especially as a fraction of the DON pool. Statistics should be used to evaluate whether the decrease with depth is significant.

Minor Comments: Line 3: Specify that N:P implies NO3-:PO43-. As written it is somewhat ambiguous since the study focuses on organic nutrients.

There is some redundancy regarding pigments and cyanobacteria – this is stated at least 3 times, and is probably unnecessary.

―――――――――――――――――――――――

---

## Author Comment (AC1) · 24 Mar 2017

Dear Editor and Reviewer

Thank you for sending the reviews of our manuscript (original ms#: bg-2017-8) entitled "Strong linkages between surface and deep water dissolved organic matter in the East/Japan Sea". Our response to each point suggested by Dr. Hawkes is as follows:

Major comments:

-Page 2 line 30: There is no mention of the use of reference materials (e.g. the seawater reference material from Hansell's lab). If no references were used, this should be mentioned. Maybe the results can be compared with another study from the same waters.

[Figure]

=> We used reference materials for verifying DOC analysis. In the revised version, we stated that "The reliability of the measurements was verified on a daily basis by analysis of DOC-certified seawater samples (DSR: 44–46 uM for DOC, University of Miami). The results were in good agreement with certified DSR values (deviation: <5%)".

- Page 4 Line 15 and many other places: When values are stated as $\pm$ something, are they referring to one standard deviation? This should be stated (i.e. (mean $\pm$ SD)). If this is the case, I don't think the values are significantly different on page 4 line 15.

=> Yes, they are standard deviations. In the revised version, we stated that "Most values are expressed as the mean $\pm$ standard deviation (SD) in this study" in Materials and Methods section. The lowest concentration of DOC as well as THAA was found in the southern area of the EJS and linked to the water circulations in the EJS. We stated that "There is no notable DOC anomaly in the EJS that would indicate significant lateral inputs from rivers or other sources (Kim et al., 2015). This is consistent with the fact that there are no major rivers that drain into the EJS, which is fed exclusively by the Pacific Ocean. Abyssal circulation in the EJS has relatively strong cyclonic flows along the basin periphery and sluggish flows in the interior region (Fig. 1; Senjyu et al., 2005). The lowest concentration of DOC was found in the southern area of the EJS along the abyssal circulation, with a similar trend of AOU (apparent oxygen utilization) (Kim et al., 2015)".

-Page 4 Line 31 onwards: They talk about the reactivity of THAAs in the EJS compared with BATS and HOT, but there is no indication of how long a time passes between the sampling points in each case. A comparison of decrease in concentration alone cannot be used to talk about reactivity, without time. This needs an extra paragraph of discussion.

=> The reactivity of THAA is indicated by the C-normalized yield (%DOC as THAA). We stated that "Bioassay experiments have demonstrated that THAA DOC yields decline

with increasing biodegradation over time and are indicative of THAA reactivity (Davis and Benner 2007, Davis et al. 2009). In addition, the global ocean distribution of THAA yields is consistent with bioassay observations (Kaiser and Benner 2009)".

-Page 5 Line 31 onwards: They mention a 'significant but weak' correlation between two variables without providing any statistical data or graphs, only a table. I think this sentence should include at least an rËĘ2, p value and n.

=> In the revised version, added r2 value (0.001) and sample number (n=66).

-Page 6 lines 9-25: They state that D-amino acids are more resistant to decomposition, yet the ratio of D:L is lower in the deep water than surface water. These two things don't seem to connect, please clarify.

=> In the revised version, t-tests were done in order to compare the ratios of D:L between the surface and deep waters. We stated that "the average D:L ratios between surface layer and deep layer were not significant different (p > 0.05)". Deleted "with a slight decrease observed at depth".

-Section 3.5: I don't understand how they make the step from the reported data to the statement: "our results suggest that the shallowing or slowdown of deep-water formation in response to atmospheric warming would bring about a considerable decrease in the oceanic storage of bioavailable DOM and a consequent positive feedback in the climate system". This needs to be discussed more thoroughly.

=> In the revised version, we added a figure about the variability of sea surface temperature and air temperature from 1932 to 2009 in the East/Japan Sea in order to explain linkages between storage of bioavailable DOM and climate change. We stated that "The increasing trend in average annual sea surface temperature and average winter sea surface temperature near Vladivostok, the most northern part of the Japan Basin, was synchronized with the warming trend (2oC) of winter air temperatures (December – February) from 1932 to 2009 in the EJS (Fig. 7)".

**BGD**

General comment: There is a lot of discussion of the trends in shallow vs. deep waters, but very little discussion of the lateral data between stations (i.e. figures 2,3 and 5). Some of the stations seem to be significantly different from each other and I would like to see a little discussion on whether these changes fit with the other trends, for example degradation of L-amino acids in comparison to D- or loss of DON preferentially over DOC. What is the circulation time difference between the different stations in deep water?

=> In the revised version, we stated that "There is no notable DOC anomaly in the EJS that would indicate significant lateral inputs from rivers or other sources (Kim et al., 2015). This is consistent with the fact that there are no major rivers that drain into the EJS, which is fed exclusively by the Pacific Ocean. Abyssal circulation in the EJS has relatively strong cyclonic flows along the basin periphery and sluggish flows in the interior region (Fig. 1; Senjyu et al., 2005). The lowest concentration of DOC was found in the southern area of the EJS along the abyssal circulation, with a similar trend of AOU (apparent oxygen utilization) (Kim et al., 2015)". In addition, the DOC and DON concentrations in the deep EJS were remarkably stable, and neither systematic nor significant differences were observed among the stations. However, the THAA data provide a sensitive indicator (%DOC) of bioavailable DOC throughout the water column.

Minor comments: - Page 1 Line 26: Should this read 'surrounded by the Korean Peninsula...?' – The Authors presumably know better than I.

=> As shown in Figure 1, the East/Japan Sea is surrounded by Korea, Japan, and Russia.

- Page 1 Line 29: The Korea/Tsushima Strait is not indicated on the map, which would be useful.

=> In the revised version, indicated the Korea/Tsushima Strait in Figure 1.

- Page 2 Line 30: 'Based on colorimetric analysis' (no 'the'). Is it possible to reference the method used here?

=> In the revised version, deleted 'the'. We stated that "The reliability of the measurements was verified on a daily basis by analysis of DIN and DOC-certified seawater samples (MOOS-1: 23.7±0.9 uM for DIN, National Research Council; DSR: 44–46 uM for DOC, University of Miami). The results were in good agreement with certified DSR values (deviation: <5%)".

-Page 3 Line 22-23: This implicitly assumes that all organic matter in these deep waters comes from surface waters. Can this be substantiated?

=> According to Kim et al. (2015, Scientific Reports), the $\delta$ 13C-DOC in the East/Japan Sea (EJS) ranged from $-$ 20.4 to $-$ 21.7% (avg.: $-$ 21.3 $\pm$ 0.4%) and showed constant values in deep waters. This result indicated that the source of DOC in the deep EJS is found to be of marine origin. The DOC and DON concentrations in the deep EJS were remarkably stable, and neither systematic nor significant differences were observed among the stations.

-Page 4 Lines 10-11: Amino acids are stated in uM, I think this should be nM.

=> corrected in the revised version.

-Page 4 Line 24: I don't think the 'normalized yield of THAA' is a commonly understood concept, so it should be briefly explained here.

=> In the revised version, we stated that "The DOC- and DON-normalized yield of THAA was calculated as the percentage of DOC and DON measured as amino acids". The formula for calculating the THAA yield could be shown in the Materials and Methods.
* * *
[Figure]

[Figure]

[Figure]

**Fig. 1.** Figure 1

[Figure]

**Fig. 2.** Figure 7

---

## Author Comment (AC2) · 24 Mar 2017

Dear Editor and Reviewer Thank you for sending the reviews of our manuscript (original ms#: bg-2017-8) entitled "Strong linkages between surface and deep water dissolved organic matter in the East/Japan Sea". Our response to each point suggested by the second reviewer is as follows:

Major comments:

There are no definition of THAA, DOC or DON in a explicit way. DOC and DON are maybe more obvious, but a short sentence with the main meaning of THAA would be appreciated.

=> In the revised version, we stated that "Filtered (0.7 $\mu$m) water samples were subjected to acid hydrolysis for determination of the total dissolved amino acid concentration and composition".

Material and Methods should include info about error of the measurements. Same for the Tables 1 and 2. Certified Reference Materials are widely used nowadays for measurement validation, if they were not used should be said.

=> In the revised version, we stated that "Most values are expressed as the mean ± standard deviation (SD) in this study" in Materials and Methods section.

=> We used reference materials for verifying DOC and DIN analysis. In the revised version, we stated that "The reliability of the measurements was verified on a daily basis by analysis of DIN and DOC-certified seawater samples (MOOS-1: 23.7±0.9 $\mu$M for DIN, National Research Council; DSR: 44–46 $\mu$M for DOC, University of Miami). The results were in good agreement with certified DSR values (deviation: <5%)".

Along the manuscript, decreasing concentrations are always related with utilization. What about the lateral advection? Don't should be discussed?

=> In the revised version, we stated that "There is no notable DOC anomaly in the EJS that would indicate significant lateral inputs from rivers or other sources (Kim et al., 2015). This is consistent with the fact that there are no major rivers that drain into the EJS, which is fed exclusively by the Pacific Ocean. Abyssal circulation in the EJS has relatively strong cyclonic flows along the basin periphery and sluggish flows in the interior region (Fig. 1; Senjyu et al., 2005). The lowest concentration of DOC was found in the southern area of the EJS along the abyssal circulation, with a similar trend of AOU (apparent oxygen utilization) (Kim et al., 2015)".

Specific comments:

- Fig.1 - why is not Tsushima Current (TC) in the figure? Additionally, some color differentiation would be appreciated for the map A).

=> In the revised version, indicated the Tsushima Current (TC) and changed as suggested in Figure 1.

- Page 5 - Line 32 " a significant, but weak, correlation" more info is needed

=> In the revised version, added r2 value (0.001) and sample number (n=66).

- Page 6 Line 24 - Is the slight decrease significant?

=> In the revised version, t-tests were done in order to compare the ratios of D:L between the surface and deep waters. We stated that "the average D:L ratios between surface layer and deep layer were not significant differences (p > 0.05)". Deleted "with a slight decrease observed at depth".

———————————————————————

[Figure]

[Figure]

**Fig. 1.** Figure 1

---

## Author Comment (AC3) · 24 Mar 2017

Dear Editor and Reviewer

Thank you for sending the reviews of our manuscript (original ms#: bg-2017-8) entitled "Strong linkages between surface and deep water dissolved organic matter in the East/Japan Sea". Our response to each point suggested by the third reviewer is as follows:

My most significant concern is that the concluding sentence, regarding the potential decrease in DOM sequestration in the deep ocean with decreasing rates of deep-water formation, is not supported by the observations; this study did not document rates of deep-water formation nor how DOC concentrations have changed through time. The degradation index or %THAA are not precise enough metrics to be able to draw this

conclusion.

=> Although the potential decrease in DOM sequestration in the deep ocean was not directly predictable from the atmospheric warming trend, Such a change could be associated with the instability of the interior water masses. In the revised version, we added data of sea surface temperature and air temperature from 1932 to 2009 in the East/Japan Sea in order to explain linkages between storage of bioavailable DOM and climate change. We stated that "The increasing trend in average annual sea surface temperature and average winter sea surface temperature near Vladivostok, the most northern part of the Japan Basin, was synchronized with the warming trend (2oC) of winter air temperatures (December – February) from 1932 to 2009 in the EJS (Fig. 7)".

Section 3.2, lines 23-30: It's not clear that THAA's were preferentially consumed with depth, especially as a fraction of the DON pool. Statistics should be used to evaluate whether the decrease with depth is significant.

=> In the revised version, t-tests were done in order to compare the yields of amino acids between the surface and deep waters. We stated that "DOC-normalized yields of THAA (%DOC) between the surface waters and deep waters showed a significant difference (p < 0.05), however, DON-normalized yields of THAA (%DON) presented no significant difference (p = 0.41)".

Minor Comments: Line 3: Specify that N:P implies $NO_3^-:PO_4^{3-}$. As written it is somewhat ambiguous since the study focuses on organic nutrients.

=>In the revised version, "N:P ratio" was changed to "DIN:DIP (dissolved inorganic phosphate) ratio".

There is some redundancy regarding pigments and cyanobacteria – this is stated at least 3 times, and is probably unnecessary.

=> corrected as suggested in the revised version.

[Figure]

[Figure]

**Fig. 1.** Figure 7

---

## Author Response (AR1)

Earth and Marine Science, 102 Jejudaehakno, Jeju National University, Jeju-si, Jeju-do, 690-756, Korea

Tae-Hoon Kim
TEL) +82-64-754-3433
FAX) +82-64-725-2461
Email: thkim@jejunu.ac.kr

April 13$^{th}$, 2017

RE: "**Strong linkages between surface and deep water dissolved organic matter in the East/Japan Sea**", by *Kim et al.* <original ms#: bg-2017-8>.

Dear Editor,

Thank you for sending the reviews of our manuscript (original ms#: bg-2017-8) entitled "Strong linkages between surface and deep water dissolved organic matter in the East/Japan Sea". We submit the second revised version of this manuscript which takes into account all reviewers and editor's comments.

With these revisions, we hope that this paper is acceptable for publication in *Biogeosciences* at this time.

Yours sincerely,

Tae-Hoon Kim

**Reviewer #1:**
This article provides new bulk concentration data on amino acids, DOC and DON in the East/Japan Sea. The results are compared with the BATS and HOT oceanographic stations, which provides a useful reference for the relative levels of the various analytes measured. The data support recent findings that suggest that DOM is transported to the deep EJS in higher concentrations than is oceanographically typical, which is an interesting result and makes this a fascinating study region. The data are of a good quality and the discussion is appropriate. Some aspects need to be clarified before publication in Biogeosciences.

Major comments:

-Page 2 line 30: There is no mention of the use of reference materials (e.g. the seawater reference material from Hansell's lab). If no references were used, this should be mentioned. Maybe the results can be compared with another study from the same waters.
=> We used reference materials for verifying DOC analysis. In the revised version, we stated that "The reliability of the measurements was verified on a daily basis by analysis of DOC-certified seawater samples (DSR: 44–46 μM for DOC, University of Miami). The results were in good agreement with certified DSR values (deviation: <5%)" (Page 2 line 34-page 3 line 2).

- Page 4 Line 15 and many other places: When values are stated as ± something, are they referring to one standard deviation? This should be stated (i.e. (mean ± SD)). If this is the case, I don't think the values are significantly different on page 4 line 15.
=> Yes, they are standard deviations. In the revised version, we stated that "Most values are expressed as the mean ± standard deviation (SD) in this study" in Materials and Methods section (Page 3 lines 25-26). The lowest concentration of DOC as well as TDAA was found in the southern area of the EJS and linked to the water circulations in the EJS. We stated that "There is no notable DOC anomaly in the EJS that would indicate significant lateral inputs from rivers or other sources (Kim et al., 2015). This is consistent with the fact that there are no major rivers that drain into the EJS, which is fed exclusively by the Pacific Ocean. Abyssal circulation in the EJS has relatively strong cyclonic flows along the basin periphery and sluggish flows in the interior region (Fig. 1; Senjyu et al., 2005). The lowest concentration of DOC was found in the southern area of the EJS along the abyssal circulation, with a similar trend of AOU (apparent oxygen utilization) (Kim et al., 2015)" (Page 4 lines 28-32).

-Page 4 Line 31 onwards: They talk about the reactivity of THAAs in the EJS compared with BATS and HOT, but there is no indication of how long a time passes between the sampling points in each case. A comparison of decrease in concentration alone cannot be used to talk about reactivity, without time. This needs an extra paragraph of discussion.
=> The reactivity of TDAA is indicated by the C-normalized yield (%DOC as TDAA). We stated that "Bioassay experiments have demonstrated that TDAA DOC yields decline with increasing biodegradation over time and are indicative of TDAA reactivity (Davis and Benner 2007, Davis et al. 2009). In addition, the global ocean distribution of TDAA yields is consistent with bioassay observations (Kaiser and Benner 2009)" (Page 5 lines 7-10).

-Page 5 Line 31 onwards: They mention a 'significant but weak' correlation between two variables without providing any statistical data or graphs, only a table. I think this sentence should include at least an r^2, p value and n.
=> In the revised version, added *r* value (0.134) and sample number (*n*=66) (Page 6 line 21).

-Page 6 lines 9-25: They state that D-amino acids are more resistant to decomposition, yet the ratio of D:L is lower in the deep water than surface water. These two things don't seem to connect, please clarify.
=> In the revised version, t-tests were done in order to compare the ratios of D:L between the surface and deep waters. We stated that "the average D:L ratios between surface layer and deep layer were not significant different (*p* > 0.05)" (Page 7 lines 11-13). Deleted "with a slight decrease observed at depth".

-Section 3.5: I don't understand how they make the step from the reported data to the statement: "our results suggest that the shallowing or slowdown of deep-water formation in response to atmospheric

warming would bring about a considerable decrease in the oceanic storage of bioavailable DOM and a consequent positive feedback in the climate system". This needs to be discussed more thoroughly.

=> In the revised version, we added a figure about the variability of sea surface temperature and air temperature from 1932 to 2009 in the East/Japan Sea in order to explain linkages between storage of bioavailable DOM and climate change. We stated that "A trend of increasing average annual sea surface temperature and average winter sea surface temperature near Vladivostok, the most northern part of the Japan Basin, is synchronized with a warming trend ($2^o$C) in winter air temperatures (December – February) from 1932 to 2009 in the EJS (Fig. 7)" (Page 7 lines 32-34).

General comment: There is a lot of discussion of the trends in shallow vs. deep waters, but very little discussion of the lateral data between stations (i.e. figures 2,3 and 5). Some of the stations seem to be significantly different from each other and I would like to see a little discussion on whether these changes fit with the other trends, for example degradation of L-amino acids in comparison to D- or loss of DON preferentially over DOC. What is the circulation time difference between the different stations in deep water?

=> In the revised version, we stated that "There is no notable DOC anomaly in the EJS that would indicate significant lateral inputs from rivers or other sources (Kim et al., 2015). This is consistent with the fact that there are no major rivers that drain into the EJS, which is fed exclusively by the Pacific Ocean. Abyssal circulation in the EJS has relatively strong cyclonic flows along the basin periphery and sluggish flows in the interior region (Fig. 1; Senjyu et al., 2005). The lowest concentration of DOC was found in the southern area of the EJS along the abyssal circulation, with a similar trend of AOU (apparent oxygen utilization) (Kim et al., 2015)" (Page 4 lines 28-32). In addition, the DOC and DON concentrations in the deep EJS were remarkably stable, and neither systematic nor significant differences were observed among the stations. However, the TDAA data provide a sensitive indicator (%DOC) of bioavailable DOC throughout the water column.

Minor comments:
- Page 1 Line 26: Should this read 'surrounded by the Korean Peninsula...?' – The Authors presumably know better than I.

=> As shown in Figure 1, the East/Japan Sea is surrounded by Korea, Japan, and Russia.

- Page 1 Line 29: The Korea/Tsushima Strait is not indicated on the map, which would be useful.

=> In the revised version, indicated the Korea/Tsushima Strait in Figure 1.

- Page 2 Line 30: 'Based on colorimetric analysis' (no 'the'). Is it possible to reference the method used here?

=> In the revised version, deleted 'the'. We stated that "The reliability of the measurements was verified on a daily basis by analysis of DIN and DOC-certified seawater samples (MOOS-1: 23.7±0.9 μM for DIN, National Research Council; DSR: 44–46 μM for DOC, University of Miami). The results were in good agreement with certified DSR values (deviation: <5%)" (Page 2 line 34-page 3 line 2).

-Page 3 Line 22-23: This implicitly assumes that all organic matter in these deep waters comes from surface waters. Can this be substantiated?

=> According to Kim et al. (2015, Scientific Reports), the δ 13C-DOC in the East/Japan Sea (EJS) ranged from − 20.4 to − 21.7‰ (avg.: − 21.3 ± 0.4‰) and showed constant values in deep waters. This result indicated that the source of DOC in the deep EJS is found to be of marine origin. The DOC and DON concentrations in the deep EJS were remarkably stable, and neither systematic nor significant differences were observed among the stations.

-Page 4 Lines 10-11: Amino acids are stated in μM, I think this should be nM.
=> corrected in the revised version.

-Page 4 Line 24: I don't think the 'normalized yield of THAA' is a commonly understood concept, so it should be briefly explained here.
=> In the revised version, we stated that "The DOC- and DON-normalized yield of TDAA was expressed as a percentage of the total DOC and DON". The formula for calculating the TDAA yield is provided in the

Materials and Methods (Page 3 lines 14-19).

**Reviewer #2:**
The manuscript by Kim et al. describes concentrations of dissolved organic nitrogen, dissolved organic carbon and amino acids in the East Japan Sea. A reasonable amount of new data is presented and the presentation is generally aceptable. The manuscript is well-written, and the organization is easy and clear to follow. It is a short paper that fits within the scope of the journal. The topic is relevant given the possible effects of changes in oceanic carbon chemistry on bacterioplankton communities and subsequent biogeochemical nutrient cycles. The article suggest an interesting link between microbial metabolism and oceanographic processes, it contributes to improve our understanding on the carbon cycle of one of the most understudied variables, so I consider that this manuscript would be suitable for publication after minor revisions.

Major comments:

There are no definition of THAA, DOC or DON in a explicit way. DOC and DON are maybe more obvious, but a short sentence with the main meaning of THAA would be appreciated.
=> In the revised version, we stated that "Filtered (0.7 µm) water samples were subjected to acid hydrolysis for determination of the total dissolved amino acid concentration and composition. The TDAA concentrations, including both free amino acids and combined amino acids, were measured using high-performance liquid chromatography (Agilent 1260 with fluorescence detector)" (Page 3 lines 4-6).

Material and Methods should include info about error of the measurements. Same for the Tables 1 and 2. Certified Reference Materials are widely used nowadays for measurement validation, if they were not used should be said.
=> In the revised version, we stated that "Most values are expressed as the mean ± standard deviation (SD) in this study" in Materials and Methods section (Page 3 lines 25-26).
=> We used reference materials for verifying DOC and DIN analysis. In the revised version, we stated that "The reliability of the measurements was verified on a daily basis by analysis of DIN and DOC-certified seawater samples (MOOS-1: 23.7±0.9 µM for DIN, National Research Council; DSR: 44–46 µM for DOC, University of Miami). The results were in good agreement with certified DSR values (deviation: <5%)" (Page 2 line 34-page 3 line 2).

Along the manuscript, decreasing concentrations are always related with utilization. What about the lateral advection? Don't should be discussed?
=> In the revised version, we stated that "There is no notable DOC anomaly in the EJS that would indicate significant lateral inputs from rivers or other sources (Kim et al., 2015). This is consistent with the fact that there are no major rivers that drain into the EJS, which is fed exclusively by the Pacific Ocean. Abyssal circulation in the EJS has relatively strong cyclonic flows along the basin periphery and sluggish flows in the interior region (Fig. 1; Senjyu et al., 2005). The lowest concentration of DOC was found in the southern area of the EJS along the abyssal circulation, with a similar trend of AOU (apparent oxygen utilization) (Kim et al., 2015)" (Page 4 lines 28-32).

Specific comments:

- Fig.1 - why is not Tsushima Current (TC) in the figure? Additionally, some color differentiation would be appreciated for the map A).
=> In the revised version, indicated the Tsushima Current (TC) and changed as suggested in Figure 1.

- Page 5 - Line 32 " a significant, but weak, correlation" more info is needed
=> In the revised version, added $r$ value (0.134) and sample number ($n$=66) (Page 6 line 21).

- Page 6 Line 24 - Is the slight decrease significant?

=> In the revised version, t-tests were done in order to compare the ratios of D:L between the surface and deep waters. We stated that "the average D:L ratios between surface layer and deep layer were not significant differences ($p > 0.05$)" (Page 7 lines 11-13). Deleted "with a slight decrease observed at depth".

**Reviewer #3:**

Major Comments: This study documents dissolved organic carbon (DOC) and total hydrolysable amino acid (THAA) concentrations in the East Japan Sea, and compares these with values from the North Atlantic and North Pacific. The East Japan Sea is a region of deep water formation, and the study describes elevated THAA concentrations in deep waters that they interpret result from export from surface waters. They also assume that the export of THAAs from surface to depth as a source of bioavailable DOM to deep waters. The study also examines ratios of amino acids to calculate a degradation index proxy for degradation of dissolved organic matter. They find little to no increase in the degradation index with depth, and the absolute degradation index is intermediate between values observed at the HOT and BATS locations. The manuscript is clear and straightforward. The data are largely consistent with prior work on marine DOM, and the conclusions largely follow from the data.

My most significant concern is that the concluding sentence, regarding the potential decrease in DOM sequestration in the deep ocean with decreasing rates of deep-water formation, is not supported by the observations; this study did not document rates of deep-water formation nor how DOC concentrations have changed through time. The degradation index or %THAA are not precise enough metrics to be able to draw this conclusion.
=> Although the potential decrease in DOM sequestration in the deep ocean was not directly predictable from the atmospheric warming trend, such a change could be associated with the instability of the interior water masses. In the revised version, we added data of sea surface temperature and air temperature from 1932 to 2009 in the East/Japan Sea in order to explain linkages between storage of bioavailable DOM and climate change. We stated that "A trend of increasing average annual sea surface temperature and average winter sea surface temperature near Vladivostok, the most northern part of the Japan Basin, is synchronized with a warming trend ($2^{\circ}$C) in winter air temperatures (December – February) from 1932 to 2009 in the EJS (Fig. 7)" (Page 7 lines 32-34).

Section 3.2, lines 23-30: It's not clear that THAA's were preferentially consumed with depth, especially as a fraction of the DON pool. Statistics should be used to evaluate whether the decrease with depth is significant.
=> In the revised version, t-tests were done in order to compare the yields of amino acids between the surface and deep waters. We stated that "DOC-normalized yields of TDAA (%DOC) between the surface waters and deep waters showed a significant difference ($p < 0.05$), however, DON-normalized yields of TDAA (%DON) presented no significant difference ($p = 0.41$)" (Page 5 lines 12-14).

Minor Comments: Line 3: Specify that N:P implies NO3-:PO43-. As written it is somewhat ambiguous since the study focuses on organic nutrients.
=>In the revised version, "N:P ratio" was changed to "DIN:DIP (dissolved inorganic phosphate) ratio" (Page 4 line 17).

There is some redundancy regarding pigments and cyanobacteria – this is stated at least 3 times, and is probably unnecessary.
=> corrected as suggested in the revised version.

**Associate Editor:**

Thanks for providing responses to reviewers of your article "Strong linkages between surface and deep water dissolved organic matter in the East/Japan Sea" by Tae-Hoon Kim, Guebuem Kim, Yuan Shen, and Ronald Benner. Thanks for providing responses to three Reviewers of your BG discussion paper (bg-2016-222). I would like to invite you to submit a revised version of the article based on your responses, and considering specially the following issues:

1. Abstract: "These observations suggest that the transport of bioavailable DOM to microbial food webs in deep waters of the EJS is sensitive to changes in deep-water renewal rates. "

Editor: This is not clear from the information provided in the Abstract section, and is also a concern to Reviewer 3for which I ask you to consider it in detail.

=> In the revised version, we stated that "Climate warming during the past few decades in the EJS is weakening deep convection during the winter, and one consequence of this reduction in deep convection is a decline in the supply of bioavailable DOM from surface waters" (Page 1 lines 21-23).

2. R1. "Page 5 Line 31 onwards: They mention a 'significant but weak' correlation between two variables without providing any statistical data or graphs, only a table. I think this sentence should include at least an r2, p value and n. => In the revised version, added r2 value (0.001) and sample number (n=66).

Editor: Please note that regression (r2) considers independent (X) and dependent (Y) variables, whereas in correlation (r) implies that there are no dependent variables. Use accordingly.

=> In the revised version, added *r* value (0.134) and sample number (*n*=66) (Page 6 line 21).

3. R2. "There are no definition of THAA, DOC or DON in a explicit way. DOC and DON are maybe more obvious, but a short sentence with the main meaning of THAA would be appreciated. "

In the revised version, we stated that "Filtered (0.7 μm) water samples were subjected to acid hydrolysis for determination of the total dissolved amino acid concentration and composition".

Editor: Even though there is no consensus on this, consider the use of DFAA (Dissolved free amino acids) instead of THAA since it would be more clear for readers.

=> We now use TDAA throughout the revised manuscript to improve clarity. We stated that "Filtered (0.7 μm) water samples were subjected to acid hydrolysis for determination of the total dissolved amino acid concentration and composition. The TDAA concentrations, including both free amino acids and combined amino acids, were measured using high-performance liquid chromatography (Agilent 1260 with fluorescence detector)" (Page 3 lines 4-6).

4. DOC and DON normalized yields of amino acids, should be AA-N/DON and AA-C/DOC, not %DON or %DOC.

=> clarified definitions of DOC and DON normalized yields of amino acids in the revised version. DOC- and DON-normalized amino acids TDAA are expressed as a percentage of the total DOC and DON (TDAA (%DOC) = 100 x TDAA-C/DOC and TDAA (%DON) = 100 x TDAA-N/DON). The formula for calculating the TDAA yield could be shown in the Materials and Methods (Page 3 lines 14-19).

5. There are too many non-standard abbreviations that if avoided it will make the article easier to read and understand (e.g. UB, TC, YB).

=> corrected as suggested in the revised version.